



# Measurement uncertainties of scanning microwave radiometers and their influence on temperature profiling

Tobias Böck, Bernhard Pospichal, Ulrich Löhnert

Institute for Geophysics and Meteorology, University of Cologne, Cologne, 50923, Germany

*Correspondence to*: tobias.boeck@uni-koeln.de

**Abstract.** In order to improve observations of the atmospheric boundary layer (ABL), Europe's network of meteorological and hydrological services EUMETNET as well as the European Research Infrastructure ACTRIS are currently working on building networks of microwave radiometers (MWRs). Elevation-scanning MWRs are well suited to obtain temperature profiles of the atmosphere, especially within the ABL.

Understanding and assessing measurement uncertainties of scanning MWRs is therefore crucial for accurate temperature profiling. In this paper we discuss measurement uncertainties due to the instrument set-up and originating from external sources, namely (1) horizontal inhomogeneities of the atmosphere, (2) pointing errors or a tilt of the instrument, (3) physical obstacles in line of sight of the instrument, and (4) radio frequency interference (RFI). Horizontal inhomogeneities from observations at the Jülich Observatory

of Cloud Evolution (JOYCE) are shown to have a small impact on retrieved temperature profiles (< |0.22 K| in the 25th/75th percentiles below 3000 m). Typical instrument tilts, that could be caused by uncertainties during the instrument set-up, also have a very small impact on temperature profiles and are smaller than 0.1 K below 3000 m for up to 1° of tilt. Physical obstacles at ambient temperatures and in line of sight and filling the complete beam of the MWR at the lowest elevation angle of 5.4° have to be at least 600 m away

from the instrument in order to have an impact of less than 0.1 K on obtained temperature profiles. If the obstacle is 5 K warmer than its surroundings then the obstacle should be at least 2700 m away. Finally, we present an approach on how to detect RFI with an MWR with azimuth and elevation scanning capabilities. In this study we detect RFIs in a water vapor channel that does not influence the temperature retrieval, but would be relevant if the MWR were used to detect horizontal humidity inhomogeneities.

1.  Introduction and motivation

The atmospheric boundary layer (ABL) is a crucial, yet often under-sampled part of the atmosphere. ABL monitoring is important for short-range forecasting of severe weather. Top-priority atmospheric variables for NWP applications like temperature (T) and humidity (H) profiles are currently not adequately measured (Teixeira et al., 2021). Ground-based microwave radiometers (MWRs) like HATPRO (Humidity And

Temperature PROfiler) are well suited to obtain such T-profiles in the ABL as well as coarse resolution H-profiles because they provide continuous and unattended observations in nearly all weather conditions (Rose et al., 2005; Cimini et al., 2011; Löhnert & Maier, 2012). Besides zenith observations which provide path integrated values like integrated water vapor (IWV) and liquid water path (LWP) with a high temporal resolution (up to 1 s), elevation scans are used to retrieve more precise temperature profiles in the ABL

(Crewell & Löhnert, 2007), as well as to assess horizontal inhomogeneities in water vapor and cloud coverage (Marke et al., 2016). It has been shown by previous studies that the assimilation of MWR observations





is beneficial for NWP models, however such observations are not yet routinely assimilated into any operational NWP model (De Angelis et al., 2016; Caumont et al., 2016).

Building an operational network of MWRs is important to improve meteorological observations and is currently a goal which is pursued by several initiatives. The EU COST Action CA 18235 PROBE[1] (PROfiling the atmospheric Boundary layer at European scale) and the European Research Infrastructure for the observation of Aerosol, Clouds, and Trace gases (ACTRIS[2]) currently focus on establishing continent-wide quality and observation standards for MWR networks for research as well as for NWP applications. Also, driven by the E-PROFILE[3] program, a business case proposal was recently accepted by EUMETNET[4] to continuously provide MWR data to the European meteorological services (Rüfenacht et al., 2021). The German Weather Service also investigates the potential of MWR networks for improving short-term weather forecasts over Germany.

For all that it is important to assess the uncertainties of MWRs, such as biases, drifts, random noise and calibration errors (Crewell & Löhnert, 2003; Maschwitz et al., 2013; Küchler et al., 2016), but also measurement uncertainties by external sources which are the topic of this study. When installing a MWR, it has to be kept in mind that instrument setup, physical obstacles and radio frequency interference (RFI) can have an impact on observations and the quality of the obtained atmospheric profiles. Therefore, identifying and coping with these kinds of errors is one important part of the quality control, especially while searching for a suitable measurement location with minimum disturbances. E.g. if physical obstacles like trees, towers, masts, walls and also mountains are too close to the MWR they can have significant repercussions in elevation scans, which are necessary for deriving accurate T-profiles (see Figure 1 in Section 3.2 for more details on the benefit of elevation scans). That is why it is crucial to pinpoint the exact location of these obstacles and to determine a minimum distance at which they do not interfere with the MWR anymore.

This study will first introduce the HATPRO MWR in short in Section 2 and then describe the used methods like the forward model and retrieval model in Section 3, which are needed to assess measurements uncertainties. The main part of Section 4 will present a sensitivity study which uses a line-by-line radiative transfer (RT) model in which obstacles at any distance from the profiler can be simulated. Focusing on the V-band (50–60 GHz frequency range employed for temperature profiling), output comparisons with and without these simulated obstacles provide a theoretical atmospheric penetration depth per frequency channel and elevation. The impact of obstacles on T-profile retrievals will be shown as well. Other measurement uncertainties – like horizontal inhomogeneities in the ABL, pointing errors or tilts of the radiometer, and examples of RFI – and their ramifications will also be presented and analyzed.

## 2. HATPRO microwave radiometer

The instruments used in this study are HATPROs (Humidity And Temperature PROfiler) and are the most widely used MWRs within Europe[5] and are manufactured by Radiometer Physics GmbH (RPG) in Germany. These passive ground-based microwave radiometers operate within the K-band and V-band spectra (the

---

[1] https://www.cost.eu/actions/CA18235/
[2] The Aerosol, Clouds and Trace Gases Research Infrastructure - www.actris.net
[3] EUMETNET Profiling Program
[4] European Meteorological Network - https://www.eumetnet.eu/
[5] see Cloudnet: https://docs.cloudnet.fmi.fi/api/data-upload.html and https://instrumentdb.out.ocp.fmi.fi/



22–32 GHz water vapor absorption line and the 51–58 GHz oxygen absorption complex) and are among the best to obtain T-profiles and coarse H-profiles of the troposphere. HATPROs measure microwave radiances expressed as brightness temperatures (TB) in 14 different frequency channels (see Table 1) in parallel, in zenith and other elevation angles with a temporal resolution on the order of seconds. The TBs can be used to retrieve the T-profiles and H-profiles, but also path integrated values like integrated water vapor (IWV) and liquid water path (LWP) with uncertainties below 0.5 kg m$^{-2}$ and 20 g m$^{-2}$, respectively (Löhnert & Crewell, 2003). The quality of these retrievals naturally depends on how well they were trained and implemented. Retrieving T-profiles within the ABL works best when using elevation scans. That is why HATPROs measure at multiple elevations, usually using between 6 and 10 different angles between 0° and 90°. Elevations scans, although conducted with all frequency channels, are usually only used within the V-band, as the channels there are optically thick, especially channels 10–14. That means that penetration depth in these channels is low enough to ensure measurement benefits for resolving the temperature profile when using different elevation angles (see Section 4.3 and Table 2 for more details on penetration depths or maximum detection distances).

Table 1: Center frequencies, bandwidths and half-power beam widths (HPBWs) of the 14 HATPRO channels (RPG, 2015).

|  | channel # | center frequencies (GHz) | bandwidth (MHz) | half-power antenna beam width (°) |
|---|---|---|---|---|
| K-band (water vapor) | 1 | 22.24 | 230 | 3.9 – 4.6 |
|  | 2 | 23.04 | 230 |  |
|  | 3 | 23.84 | 230 |  |
|  | 4 | 25.44 | 230 |  |
|  | 5 | 26.24 | 230 |  |
|  | 6 | 27.84 | 230 |  |
|  | 7 | 31.40 | 230 |  |
| V-band (oxygen) | 8 | 51.26 | 230 | 1.8 – 2.2 |
|  | 9 | 52.28 | 230 |  |
|  | 10 | 53.86 | 230 |  |
|  | 11 | 54.94 | 230 |  |
|  | 12 | 56.66 | 600 |  |
|  | 13 | 57.30 | 1000 |  |
|  | 14 | 58.00 | 2000 |  |

TB accuracy is different for each channel and is usually below 0.5 K for Generation 5 HATPROs, according to the manufacturer RPG. This accuracy mainly consists of these 4 instrument errors which are not topic of this study: calibration repeatability, radiometric noise, drift and bias (Crewell & Löhnert, 2003; Maschwitz et al., 2013; Küchler et al., 2016). Other radiometer characteristics like antenna beam width and receiver bandwidth (but also atmospheric propagation) can also have an impact on scanning MWR measurements, depending on the frequency channel and elevation angle. However, within the V-band and the elevation angles used in this study (> 4°), these impacts are negligibly small (< 0.05 K) (Meunier et al. 2013), also considering that the half-power antenna beam width (HPBW) of a HATPRO within the V-band is only up to 2.2° (RPG, 2015) and the penetration depths of the V-band channels are limited. Additional sources of uncertainty for HATPRO products are due to radiative transfer model and absorption coefficient errors (see Section 3).





### 3. Methods

Here we describe the most important methods used for analyzing measurement uncertainties and their influence on T-profiling, especially the methods needed for the simulation of pointing errors and the simulation of physical obstacles. Measurement setups required for measuring horizontal inhomogeneities and identifying RFIs, as well as finer details in the simulations of obstacles and pointing errors are described in the results section (Section 4) as needed.

### 3.1. Forward model

In this study, non-scattering radiative transfer simulations are carried out based on Simmer (1994). Gaseous absorption is calculated according to Rosenkranz (1998), whereby the water vapor continuum is modified according to Turner et al. (2009) and the 22 GHz water vapor line width is modified according to Liljegren et al. (2005). This model depicts the energy transfer in form of electromagnetic radiation through the atmosphere via absorption and emission by gases and hydrometeors and it is modified here to simulate physical obstacles at different distances and elevation angles and to assess pointing errors.

The input to the model consists of radiosonde profile data from the Richard-Aßmann-Observatory (RAO) in Lindenberg, Germany from the year 2000 which provide pressure, height, temperature and relative humidity. There were 4 soundings per day and in total 1436 usable soundings for the year 2000. The provided height levels from the atmospheric soundings have been linearly interpolated to a spacing of 1 m below 150 m, 10 m below 10 km, and 1000 m below 30 km. When there was no data available above certain heights (mostly between 10 km and 30 km), the gaps were filled with data from the International Standard Atmosphere (ISA) at mid-latitude[6]. The forward model uses the same height levels as the interpolated soundings. Input parameters include the 14 HATPRO frequency channels and the used elevation angles (5.4°, 10.2°, 19.2°, 30°, 42° and 90°). For the most part of this study, the focus lies on the 7 V-band channels, as only these are needed to later retrieve T-profiles (see Section 3.2)

Within this non-scattering RT model, the calculation of the optical thickness $\tau$ of the atmosphere plays an important role. It is dimensionless and calculated via the integration of the gas absorption coefficient $\beta_a$ over the optical path $s'$:

$$\tau(s, \nu) = \int_0^s \beta_a (s', \nu) \, ds'. \tag{1}$$

$\tau$ is not only dependent on the total optical path $s$, but also on the frequency $\nu$. First, $\tau$ is calculated in zenith direction. By multiplying the zenith $\tau$ by $1/\cos(\theta)$, with $\theta$ being the zenith angle, we arrive at the optical depth at any arbitrary elevation angle assuming horizontal homogeneity. In order to simulate physical obstacles, the optical thickness $\tau$ can be manually set to very high values (e.g. $\tau = 100$) at the desired height level and in turn the desired distance for any elevation angle. Doing this will ensure that no radiation beyond the simulated obstacle will pass through to the radiometer position. Without further modifications the obstacles will have the ambient temperature of their surroundings derived from the radiosonde input. Within this study, the influence of "heated" obstacles with +5 K compared to ambient temperature have

---

[6] ICAO standard atmosphere: https://www.oxfordreference.com/display/10.1093/oi/authority.20110803095955775;jsessionid=032BE96B6B7C7004CEF8FBF09CC9F731



also been analyzed. Such heated obstacles could for example represent facades of buildings or mountain slopes which are lit by the sun.

For the simulation of obstacles some assumptions were made: (1) horizontal homogeneity of the atmosphere, (2) the obstacle is a perfect blackbody with ambient (or +5 K) temperature, and (3) the whole cone from the antenna HPBW is filled with the obstacle. As an example, for an antenna beam width of 3°, the

radius of the cone is about 13.1 m when 500 m away from the radiometer and about 26.2 m when 1000 m away.

As the forward model should simulate TB measurements from a MWR, radiometric white gaussian noise of 0.5 K has been added to all 7 V-band channels.

### 3.2.  Retrieval model

With the help of a retrieval model or algorithm, we can calculate a response of retrieved atmospheric parameters like T-profiles to the TB measurement uncertainties from obstacles, pointing errors or horizontal inhomogeneities. The temperature retrieval itself is trained and tested with years of radiosonde data from a certain location; in our case from near or the same locations where the measured (or simulated) TBs are gained from. In the analysis of horizontal inhomogeneities (Section 4.1), retrieval coefficients derived

from De Bilt in the Netherlands were utilized. De Bilt was chosen due to its climatological similarity to JOYCE in Jülich, where the measurements were conducted, and because it was the nearest place where retrieval coefficients were available. Conversely, when examining simulated pointing errors (Section 4.2) and the influence of obstacles (Section 4.3), retrieval coefficients from RAO in Lindenberg were employed. RAO in Lindenberg was selected because it provided a substantial dataset of radiosonde data spanning multiple

years, enabling statistical analysis. Additionally, these radiosonde data are already utilized as inputs in the forward model.

Regression coefficients are generated through multi-variate linear regressions. These coefficients are needed to calculate T-profiles from TBs as seen in equation 2

$$T_z(\nu, \theta) = d_0 + \sum_\nu \sum_\theta d_{1\nu\theta} \ TB_{\nu\theta}, \tag{2}$$

where $d_0$ is the offset value, $d_{1\nu\theta}$ are the regression coefficients, $z$ is the index for each output layer height,

and $\nu$ and $\theta$ stand for frequency and elevation angle (Crewell and Löhnert, 2007; Meunier et al., 2013). This T-profile retrieval takes the used frequency channels and elevation angles into account. In this study these are the channels 8–14 from the V-band and the 6 elevation angles 5.4°, 10.2°, 19.2°, 30°, 42° and 90°. However, channels 8–10 are only used at 90° elevation, while channels 11–14 are used at all 6 elevation angles. Another approach are less accurate zenith-only T-profile retrievals which only take frequency into account

and which are calculated as seen in equation 3:

$$T_z(\nu) = c_0 + \sum_\nu (c_{1\nu} \ TB_\nu + c_{2\nu} \ TB_\nu^2). \tag{3}$$

Here, $c_0$ is the temperature offset and the regression coefficients are $c_{1\nu}$ and $c_{2\nu}$. There is an added quadratic term compared to equation 2 which helps improve the retrieval accuracy for zenith only observations. Both



retrievals output temperature at 43 height levels, from the surface up to 10 km height, with a higher vertical spacing in the lower 5 km.


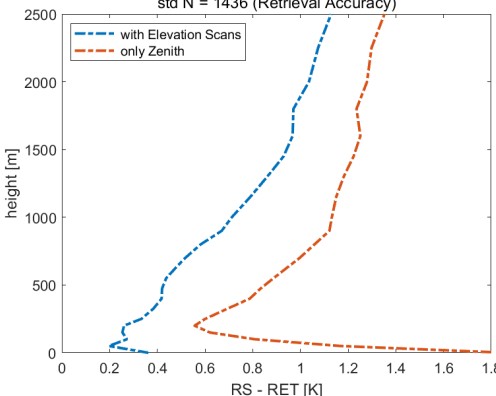

**Figure 1:** Retrieval accuracy of retrievals which make use of elevation scans (blue line, with 6 elevations angles) versus retrievals which only use zenith measurements (red line). Shown are the standard deviations (std) of the mean differences of T-profiles from radiosondes (RS) and the retrieved T-profiles (RET) from forward modeled TBs from these radiosondes (T-profiles from radiosondes minus retrieval). These were calculated from 1436 radiosondes from the year 2000 at RAO.


The improvement of temperature profile accuracy using elevation scans compared to zenith only observations can be seen in Figure 1 . Therefore it is strongly advised to use elevation scanning when possible. T-profile retrievals which make use of elevations scans have a higher accuracy within the whole lower tropo-

sphere than T-profile retrievals using only zenith measurements, especially in the lowest 200 m of the atmosphere where the difference in standard deviation is up to 1.4 K, while the reduction of the uncertainty between 200–1000 m is around 0.5 K and between 1000–5000 m around 0.2 K.

## 4. Results and discussion

Here, measurement uncertainties of scanning HATPRO microwave radiometers are shown and discussed
in detail. These include horizontal inhomogeneities of the atmosphere (measured), pointing errors caused by a tilt of the instrument (simulated), the influence of physical obstacles in the line of sight of the instrument (simulated), and examples of RFIs and how to identify them (measured).

### 4.1. Horizontal inhomogeneities

One of the assumptions for the elevation retrieval to work properly is horizontal homogeneity of the at-
mosphere. In reality however, horizontal homogeneity is not always given. In order to investigate the impact of real life horizontal inhomogeneities on retrieved T-profiles, 3 week MWR measurements from JOYCE (Löhnert et al., 2015) from August to September 2022 are analyzed. The HATPRO was measuring at multiple elevation angles (5.4°, 10.2°, 19.2°, 30°, 42° and 90°) to the north and – immediately after the completion of such an elevation scan – did the same to the south. This scan pattern was repeated roughly every 15 minutes
(the duration of one scan is roughly 2.5 minutes). To exclude the influence of clouds, clear-sky profiles were





considered by filtering out cases with a LWP > 10 g m⁻² (for about a 30 minute period before and after a scan).

### 4.1.1. TB analysis

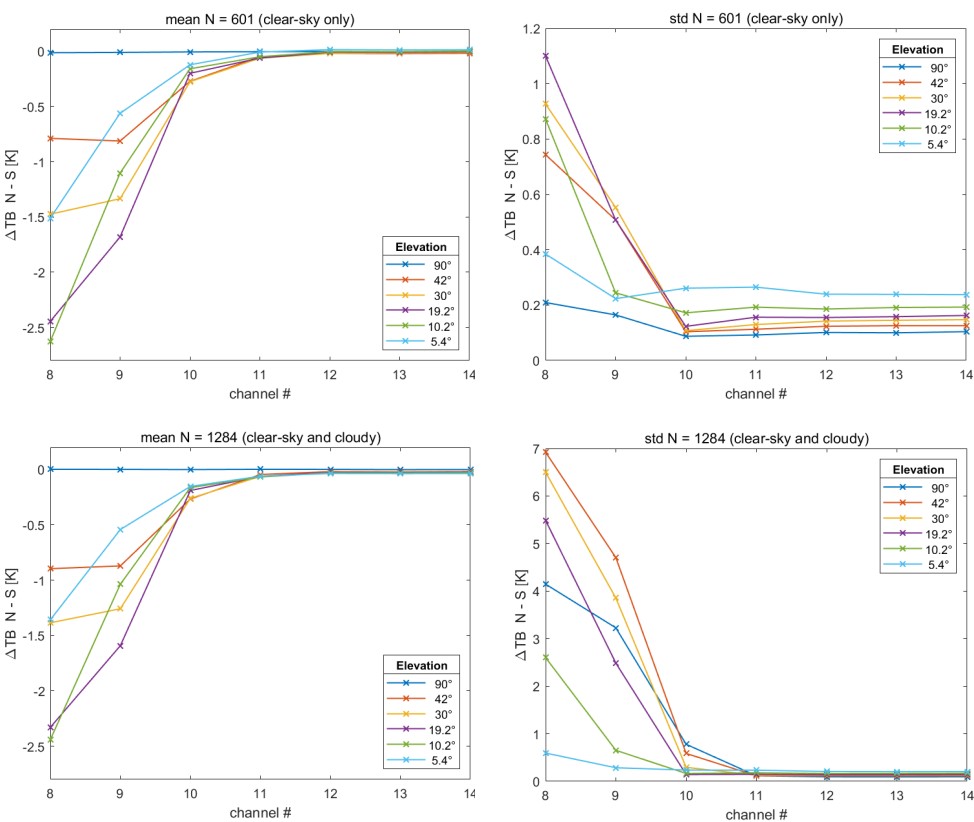


**Figure 2: TB differences from north- and south-facing radiometer measurements over a 3 week period in summer 2022 from JOYCE from clear-sky only (upper row) and clear and cloudy conditions (bottom row). On the left are the mean differences per V-band channel and elevation, on the right the corresponding standard deviations (std). Note that the y-axes for the standard deviation plots are not the same.**

For the analysis, every pair of north-facing scans and immediately following south-facing scans are com-
pared. The mean TB difference over all north- and south-facing measurement pairs as a function of V-band
channel and elevation and their standard deviation are presented in Figure 2 for clear-sky only and clear-
sky and cloudy conditions. The standard deviations – minus the corresponding radiometric noise of the TB
measurements – and thus the variability of the TB differences are an indicator of actual measured horizontal

inhomogeneities, while the mean differences, in contrast, can also be the consequence of other factors. Keep
in mind that the radiometric noise for Generation 5 HATPROs is on average 0.15 K within the V-band, ac-
cording to the manufacturer (RPG, 2015), with slightly higher values in the optically thinner channels 8–10
and slightly lower values for channels 11–14.

     Let us first focus on the optically thinnest channels of the V-band, channels 8–9. These channels reach the

deepest penetration depths and show the highest mean differences in north- and south-facing



measurements and also the highest standard deviations of the TB differences in all elevations. This is in line with expectations considering the usually highly inhomogeneous distribution of water vapor, to which channels 8–9 (and to a lesser extent channel 10) are sensitive to, along with temperature variability (Westwater et al., 2005).

Standard deviations of TB differences in channels 8 and 9 show values of up to 1.1 K at 19.2° elevation in clear-sky conditions and up to 6.9 K at 42° elevation, when cloudy conditions are included. Higher absolute standard deviations for cloudy conditions are to be expected in those channels, as horizontal inhomogeneities increase when a cloud is present in one scan but not in the corresponding scan in opposite direction. As the occurrence probability of clouds decreases towards the surface, standard deviations relative to

higher elevations are lower for lower elevations when cloudy conditions are included. The increasing optical depth in channels 8–9 at lower elevations is also a reason for these observed lower standard deviations in relation to the higher elevation angles. The standard deviation at 90° elevation in channels 8–9 can be explained by the fact that – although at zenith both north- and south-facing scans are pointing in the exact same direction – the measured variability here is not due to horizontal inhomogeneities but rather due to

the noise of the instrument and the 2.5 minute time difference between a north- and a south-facing scan, in which atmospheric variables can have changed slightly.

      The mean absolute differences in channels 8 and 9 for all the elevations vary from 0.6 K up to 2.6 K and are fairly similar for clear-sky only and clear-sky and cloudy conditions. Excluding zenith for now, the higher elevation angles in the optically thinner channels show in general lower mean absolute differences than the

intermediate elevation angles because they cover a smaller horizontal area compared to the intermediate elevation angles. The small values in mean difference in the lowest elevation can be explained by the fact that here the measurements don't reach high enough into the atmosphere and that water vapor and cloud inhomogeneities do not have much of an impact in these channels.

      As for the optically thicker channels 11–14, they show much lower mean differences and standard devia-

tions ($< 0.06 \pm 0.26$ K for clear-sky and $< 0.06 \pm 0.23$ K including cloudy conditions) than channels 8–10, as they do not penetrate the atmosphere very deeply. Lower elevations at optically thicker channels show higher standard deviations because they cover a larger horizontal area than higher elevations and are thus subject to higher near-surface temperature variability, which is also more pronounced during clear-sky conditions.

The obtained mean differences seen in Figure 2 imply that TB values in the south are consistently higher than in the north for the 3 week measurement period, independent of clear-sky or cloudy conditions. Additional forward model simulations for horizontal inhomogeneities – in which humidity profiles from RAO radiosondes were altered – reveal that an average difference of about $\Delta$IWV $\approx 3.4$ kg m$^{-2}$ in south-north direction in a clear-sky scenario would be necessary in order to achieve similar mean TB differences as seen

in Figure 2. Such a large average difference in IWV as a consequence of horizontal inhomogeneity is highly unlikely. That is why a pointing error caused by a tilt of the instrument is most probably the main reason for these observed mean differences. Via forward model simulations with different elevation angles including simulated instrument tilts, we found similar patterns as seen in Figure 2 when assuming a tilt of 0.6° in south-north direction (see Section 4.2 and Figure 4 and compare with Figure 2). While complex geography

around the measurement site and general weather conditions can also have an influence on the mean difference in measured TBs, their expected mean impact is significantly smaller as what is depicted in Figure



2. There is a slight mean diurnal cycle of the mean differences (maximum change of mean difference during night and day for channel 8 at 5.4° elevation compared to the mean differences in Figure 2 is smaller than ±0.2 K, all other channel and elevation combinations are smaller than this), which means the influence of daily variation of weather is rather low compared to the overall impact of tilt, but not absent.

### 4.1.2.   T-profile analysis

When retrieving a T-profile by means of an MWR scan, only zenith measurements are used for channels 8–10, whereas all available elevation angles (including zenith) are used for channels 11–14. That is the reason why the impact of horizontal inhomogeneities on retrieved T-profiles is rather small, even though mean TB differences in channel 8-9 can be as high as 2.6 K in our examples. This small impact can be seen in Figure 3 where the median T-profile differences retrieved from the north-facing and south-facing scans are shown. The percentiles shown in Figure 3 represent actual horizontal inhomogeneities and/or variabilities of the atmosphere, while the medians are a consequence of the assumed tilt of the instrument. The absolute median difference up to 3000 m is always below 0.08 K (0.05 K for clear-sky). The 25th and 75th percentiles do not exceed −0.20 K and 0.22 K, respectively, and only come close to these values near the surface where the variability is the highest. The 5th and 95th percentiles show a similar pattern and do not exceed −0.8 K and 0.8 K near the surface. It is evident from Figure 3 that variability near the surface is higher in clear-sky only conditions when compared to conditions including clouds. This peculiarity can be explained by two reasons. Firstly, different types of surfaces around the measurement site (e.g. concrete vs. grass and trees) heat up more differently when the sun is shining than when there are very cloudy conditions where heat is more evenly distributed near the surface. This heat is dispensed to the air near the surface where especially the optically thicker channels pick it up. Secondly, inversions near the surface during cloudy conditions are less likely than during clear-sky which also leads to less variability near the surface during cloudy conditions. Overall, the medians and 25th/75th  percentiles in all conditions lie within the expected uncertainty of HATPRO temperature profiles (Löhnert & Maier, 2012).



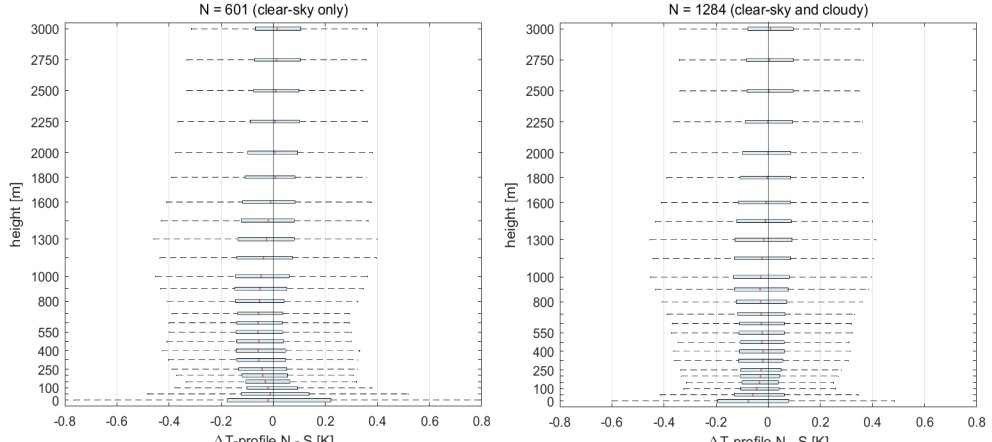

**Figure 3: Differences in derived T-profiles at certain heights from northward and southward elevation scans at JOYCE. The red lines are the medians from a 3 week period in late summer 2022, the horizontal bars show the 25th and 75th percentiles, while the dashed lines show the 5th and 95th percentiles. On the left only clear-sky conditions are shown, on the right clear and cloudy conditions.**

### 4.2.    Pointing errors or tilts

As discussed in Section 4.1, measured mean TB differences from elevation scans in opposing directions can be caused by pointing errors, not only by actual horizontal inhomogeneities of the atmosphere. Pointing errors are errors that arise from a tilt of the radiometer and are usually the result of an improper setup by the operator but can also be due to internal instrument misalignments. They impact all elevation and zenith measurements. For 30° elevation scans for example, a 1° tilt of the instrument to the south will lead to a measurement at 29° facing south and 31° facing north. Tilts have a smaller impact at higher elevations and zenith observations than on lower ones (due to trigonometric reasons). A tilt of more than 2° can have very significant repercussions on TB measurements at the lowest elevation angle of 5.4°, as the half-power antenna beam width of a HATPRO is up to 2.2° in the V-band and therefore emissions from the surface will more and more interfere the lower the elevation scans reach down.



### 4.2.1. TB analysis

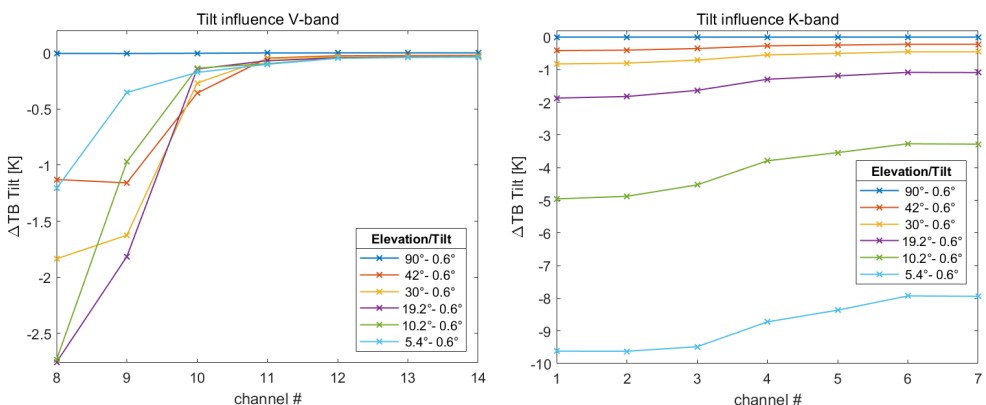

**Figure 4: Influence of tilt on TB measurements for elevation scans within the V-band (on the left) and the K-band (on the right). Depicted are simulated TB measurement differences in an inversion-free ABL from a non-tilted instrument and an instrument with 0.6° tilt (Non-tilted minus tilted).**

Figure 4 shows the simulated impact on TB measurements for different elevations when there is a pointing error of 0.6° (in opposing directions, e.g. in south-north direction). Depicted are differences of simulated

TB measurements from an MWR with and without tilt in an inversion-free atmosphere (e.g. International Standard Atmosphere ISA). Focusing on the V-band channels first, we can spot that the presence of a 0.6° pointing error yields the exact same pattern and almost the same values of ΔTBs as seen in Figure 2 on the left. From this we conclude, that the systematic differences from north- and south-facing scans in Section 4.1 are actually – for the most part – the result of a slightly misaligned MWR. Due to the fact that the inho-

mogeneities were analyzed by the difference of north- and south-facing scans, a real-world misalignment or tilt of the MWR of only 0.3° in south-north direction (e.g. such a 0.3° tilt at 30° elevation leads to 30.3° elevation in the north and to 29.7° elevation in the south, which makes a 0.6° pointing difference) would be enough to produce these particular ΔTB patterns and values as depicted in Figure 4. For the optically thinner K-band channels, a tilt can have an even larger influence on TB measurements, especially at lower ele-

vations. As far as T-profiling is concerned, K-band channels do not play a role at all and V-band channels 8–10 are only used in zenith, therefore diminishing the influence of tilts or pointing errors on T-profiling.

When trying to analyze water vapor inhomogeneities with a full azimuth scan at e.g. 30° elevation, a pointing error of 0.6° in a certain azimuth direction always has an impact of more than 0.4 K in all K-band channels (with up to 0.8 K in channel 1) when compared to measurements without tilt (see Figure 4 on the

right). Even though measurements in the K-band are not a focus in this study, it may be noteworthy that – without going into much detail – this would directly translate to an impact on IWV of about $\pm 0.24$ kg m$^{-2}$ in the direction of the tilt when retrieving along the 30° elevation path (measurements show that a 1 K TB difference in channel 1 at zenith corresponds to a roughly 0.6 kg m$^{-2}$ change in IWV). Please also note that within the K-band, TB uncertainties from receiver bandwidth and antenna beam width are also not negligi-

bly small as in the V-band. For our HATPRO set-up, they can vary between 0.1–0.3 K (Meunier et al., 2013).



### 4.2.2. T-profile analysis

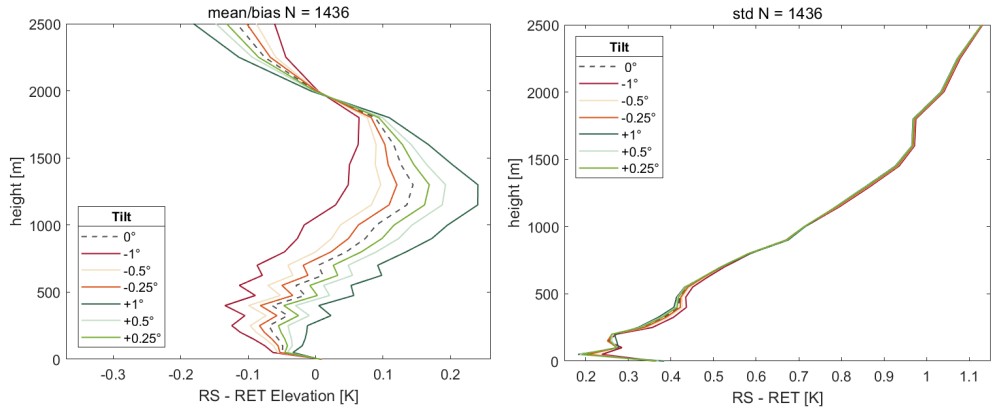

**Figure 5:** Mean differences (on the left) and their corresponding standard deviations (std, on the right) of T-profiles from radiosondes (RS) over Lindenberg and the retrieved T-profiles (RET) from forward modeled TBs from these radiosondes (T-profiles from radiosondes minus retrieval). Retrievals make use of 6 elevation angles. Shown are the influences of tilts on the radiometer of up to ±1°. The mean values are calculated from 1436 radiosondes from the year 2000 from RAO.

Figure 5 shows the impact of pointing errors of up to ±1° on T-profiling when employing elevations scans. It depicts the T-profiles mean differences (bias at 0° elevation) and their corresponding standard deviations from 1436 radiosondes from RAO over Lindenberg from the year 2000 and the retrieved profiles from simulated TBs from these radiosondes. For retrievals which make use of elevation scans, there is very little mean difference (between −0.13 K and 0.24 K) in the lowest 1000 m even when incorporating pointing errors of ±1°. Theses pointing errors hardly exceed ±0.1 K on the T-profile when compared to 0° tilt. The accuracy of these T-profile retrievals is represented by the standard deviations, on which pointing errors of ±1° have even less impact, as evident in Figure 5. Standard deviations are roughly the same for all elevations and range from 0.20–0.45 K in the lowest 500 m and 0.45–1.15 K between 500 m and 2500 m.

For retrievals which only make use of zenith measurements, the impact of pointing errors on mean TB differences and standard deviations is negligible, at least for smaller tilt angles below 2°. Nevertheless, T-profiles which make use of elevations scans have a higher accuracy than T-profiles derived from only zenith measurements, even if they are contaminated by pointing errors of up to ±1° (compare standard deviations from Figure 5 with Figure 1).



### 4.3. Influence of obstacles

When setting up a MWR at a new measurement location it has to be kept in mind that external error sources like physical obstacles (e.g. trees, buildings or nearby mountains) can have an impact on TB meas-
350    urements when they are too close and in line of sight of the MWR, especially at lower elevation angles. The impact depends on the distance, size, and temperature of the obstacle. Our simulations of such obstacles within the RT forward model have shown that in general the influence of obstacles on the measurements is the greatest within an inversion-free troposphere. The International Standard Atmosphere (ISA) can provide such an inversion-free example as an input. In order to simulate an obstacle, the optical thickness of
355    the atmosphere within the RT model can be set to a very high value at the desired distance (see Section 3.1).

#### 4.3.1. TB analysis

Figure 6 shows the impact of obstacles with ambient temperature on TB measurements within the V-band in a standard atmosphere for certain elevation angles. The difference in TB measurements with and without an obstacle tells us the exact impact of a certain obstacle. In general the conducted simulations show that
360    the impact of an obstacle is getting higher (1) the nearer an obstacle is to the MWR, (2) the higher the elevation angle is, (3) the optically thinner the frequency channel is and (4) the higher the temperature of the obstacle is compared to its surroundings.



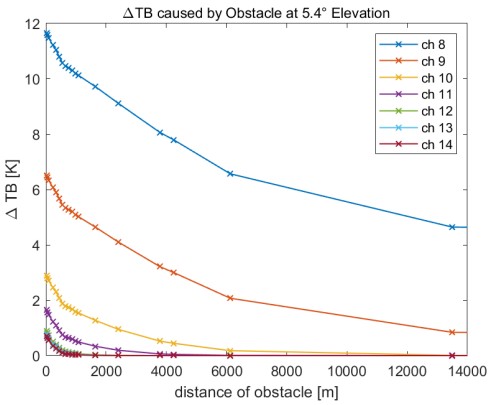

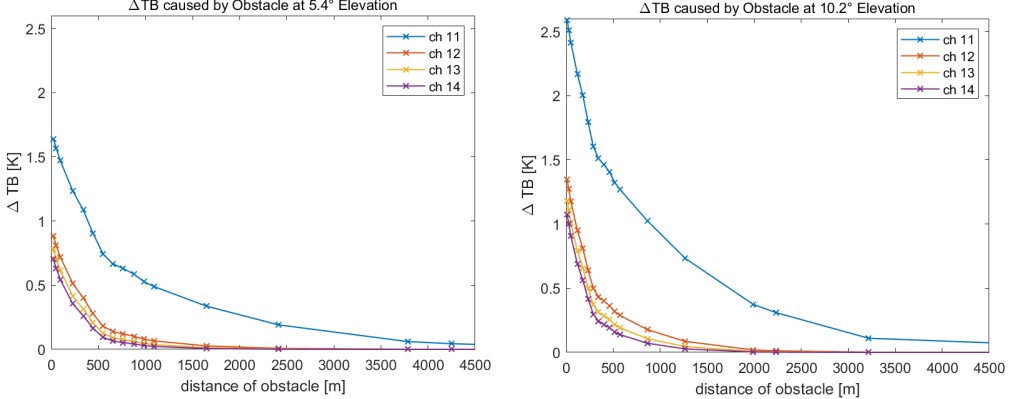

Figure 6: Impact of obstacles with ambient temperature on TB measurements at different distances from the MWR in a standard atmosphere. Shown are the differences of TBs with and without an obstacle for elevation angles 5.4° and 10.2° within the V-band. At the top for all V-band channels, at the bottom for channels 11–14.

Channels 8 and 9 – which are the optically thinnest channels within the V-band – can reach far into the atmosphere, even at low elevations. They can detect obstacles at ambient temperature at the lowest elevation angle of 5.4° from more than 30 km away. If we use a ΔTB of ≤ 0.1 K as a detection threshold, channel 10 at the lowest elevation still can detect obstacles which are more than 10 km away. But as these channels are only used in zenith for T-profile retrievals, we focus here on channels 11–14. At 5.4° elevation, channel 11 can detect ambient temperature obstacles from ~3000 m away, while channels 12–14 have a detection range of 880 m to around 500 m. At 10.2° elevation, these distances increase to ~3500 m (for channel 11) and to 1200 m or 750 m (for channel 12 – 14), respectively. If an obstacle is warmer than its surroundings (e.g. 5 K warmer), the impact on TB measurements and the detection distances will increase significantly. This maximum detection distance of obstacles is also a measure of penetration depth or how far the MWR can "see" into the atmosphere. A summary of these detection distances for different channel/angles combinations can be found in Table 2.




Table 2: Maximum detection distances of obstacles for different channel/angles combinations. A TB detection threshold of 0.1 K is used. Values are given for obstacles with ambient temperature and 5K warmer.

| DETECTION DISTANCES | elevation 5.4° | elevation 10.2° | elevation 19.2° | elevation 30° |
|---|---|---|---|---|
| ch 11  +0 K | ~3000 m | ~3500 m | ~3600 m | ~4100 m |
| ch 12  +0 K | ~880 m | ~1200 m | ~1400 m | ~1650 m |
| ch 13  +0 K | ~630 m | ~900 m | ~1100 m | ~1150 m |
| ch 14  +0 K | ~500 m | ~750 m | ~950 m | ~950 m |
| ch 11  +5 K | ~5500 m | ~6300 m | ~5000 m | ~5200 m |
| ch 12  +5 K | ~2330 m | ~2100 m | ~2350 m | ~2100 m |
| ch 13  +5 K | ~1950 m | ~1850 m | ~1650 m | ~1950 m |
| ch 14  +5 K | ~1620 m | ~1700 m | ~1550 m | ~1800 m |

During the course of a day or a year, atmospheric conditions change a lot and an inversion-free tropo-
sphere/ABL is not always present, especially during winter nights. Temperature inversions near the surface
for example can dampen the impact of obstacles, which can be derived from Figure 7.

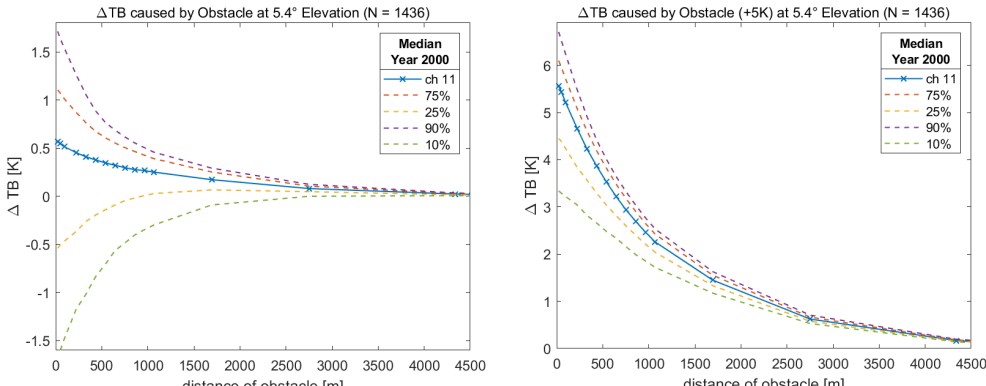

**Figure 7: Statistics of TB differences (obstacle in line of sight minus no obstacle) for channel 11 at 5.4° elevation obtained from one year of Lindenberg radiosonde data. Bold line shows the median of the differences as a function of distance to obstacle, dashed lines the corresponding percentiles. On the left obstacles have ambient temperature, on the right obstacles are 5 K warmer than their surroundings.**

Figure 7 depicts the impact of obstacles with different temperatures as a median of TB differences (ob-
stacle minus no obstacle) from Lindenberg radiosonde date for the whole year 2000 at 5.4° elevation for
channel 11. This channel was chosen as an example because it shows the highest impact from channels 11–
14. Looking at the 25[th] and 10[th] percentile lines, these indicate cases with inversions near the surface, while
the 90[th] percentile line approximately represents the inversion-free scenario as seen before in Figure 6. For
an ambient temperature obstacle, the ΔTB can even become negative when there is an inversion, meaning
that the colder obstacle near the surface blocks the MWR from observing warmer atmospheric layers above.
For an obstacle which is 5 K warmer than its surroundings, the impact on ΔTB observations is significantly
higher and will likely be positive, even when there are moderate inversions present near the surface. Over-
all, the average impact of obstacles is highest in an inversion-free atmosphere.





### 4.3.2. T-profile analysis

Previously, the focus of our discussions has been on the influence of obstacles on TB measurements. In the following, we will shift our attention to the influence of obstacles on retrieved T-profiles. Retrievals have revealed that the influence of obstacles on T-profiles is always the highest in inversion-free ABL cases. That is why the following results have been obtained in such inversion-free cases. Figure 8 shows the impact on T-profiles from different obstacles at certain distances to a MWR in a standard atmosphere. To minimize

the impact of obstacles ($\leq |0.1\,\mathrm{K}|$), such as a big tree or a nearby building, which possess ambient temperatures and are visible at an elevation of 5.4°, they must be situated at a distance of greater than 600 meters (keep in mind that at 600 m distance an obstacle would need to be at least 57 m tall in order to block the line of sight of the MWR at 5.4° elevation). In the case of larger obstacles, such as skyscrapers or nearby mountains that can be observed at elevations of 5.4° and 10.2°, the MWR must be positioned at a distance

of at least 1500 meters. For even bigger obstacles, such as high mountains, which can be seen at elevations of 5.4°, 10.2°, and 19.2°, they must be situated at a minimum distance of 2500 meters away to effectively minimize their impact on T-profiles. If these obstacles are to be 5 K warmer than their surroundings and still have an impact of $\leq |0.1\,\mathrm{K}|$ on T-profiles, these distances increase to more than 2700 m, 3500 m and 4000 m, respectively.




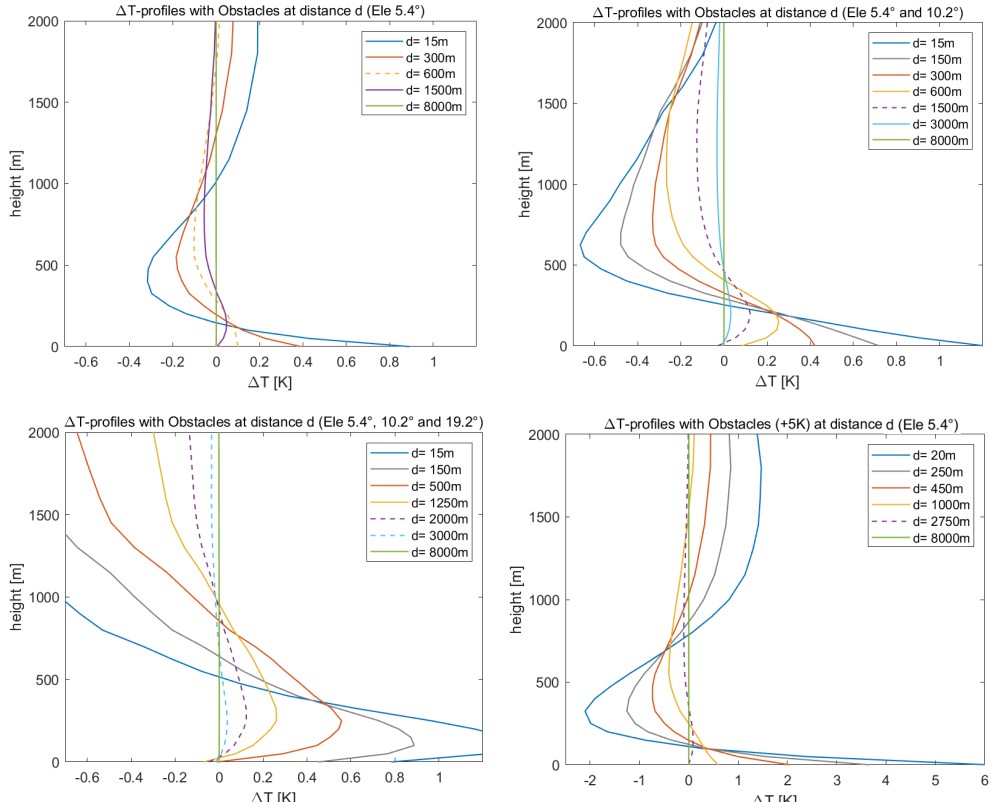

**Figure 8: Impact of different obstacles on retrieved T-profiles in a standard atmosphere. Depicted are the differences in T-profiles with and without obstacles at give distances to the MWR. The upper left shows the impact of an ambient temperature obstacle at 5.4° elevation, the upper right shows an obstacle which can be seen at 5.4° and 10.2°, while the lower left shows an obstacle which can be seen at 5.4°, 10.2° and 19.2°. The lower right shows the impact of a "heated" obstacle at different distances for 5.4° elevation which is 5 K warmer than its surroundings. Dashed lines indicate that the obstacle at this distance has an impact of ≤ 0.1 K on the T-profile.**

### 4.4. Identification of radio frequency interference (RFI)

Not only physical obstacles, pointing errors or horizontal inhomogeneities but also RFI can have repercussions on MWR measurements (Nat. Research Council, 2010). As an example, directional radio links and other telecommunication systems can be the source of such interferences, as their signal strength can be several orders of magnitudes stronger than normal atmospheric signals. In order to determine the strength and the direction of origin of interferences, full azimuth scans at several elevations are necessary. RFIs (as well as – to various degrees of accuracy – obstacles, instrument tilts and/or horizontal inhomogeneities) can be determined via the following proposed 4-step method for every HATPRO with an azimuth motor (keep in mind that interval and threshold values are not fixed and can be adjusted as seen fit): (1) Do full 360° azimuth scans, e.g. with 10° azimuth intervals at several elevations. (2) Check for clear-sky conditions. If the mean LWP of a 30 minute interval before and after one scan is below 10 g m$^{-2}$, then it is most likely clear-sky. (3) Calculate the difference of the TB measurement of one azimuth angle and the minimum of a whole azimuth scan for each azimuth angle for each channel. This is also called the scan difference here. (4) If the scan difference is greater than a defined threshold, e.g. 1 K, a disturbance has been detected.

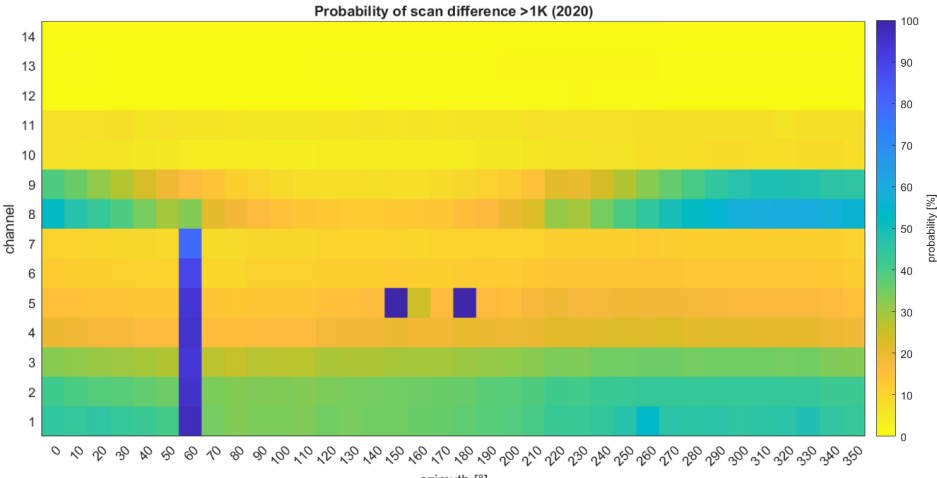

Figure 9: Mean probability of disturbance (the higher the percentage, the more pronounced the disturbance) from HAT-
PRO azimuth scans at 30° elevation for all 14 channels from 2020 at JOYCE. If the scan difference (TB measurement at
a certain azimuth angle minus the measured minimum TB of a full 360° azimuth scan) of a channel is greater than 1 K,
a disturbance has been detected. Presented are only clear-sky conditions with a total of 7390 scans.

Figure 9 shows the percentage of cases in which the scan difference for the HATPRO at JOYCE is more
than 1 K for the whole year of 2020 at 30° elevation. High values in blue indicate significant disturbances.
For channels 1–7 (maybe even channel 8) at 60° azimuth we can clearly see a lightning rod which has been
installed near the HATPRO (less than 5 m away) in 2019. This disturbance always shows up within the K-
band throughout 2020 to 2023 and is obviously a physical obstacle. As this lightning rod is very thin and
only fills out a small part of the MWR beam, it hardly shows up in the V-band. TBs measured within the V-
band are much closer to ambient temperature than in the K-band, and therefore the lightning rod at surface
temperatures does not stand out enough from the signal received from the left and right from it to have a
detectable impact on V-band channels.

Another significant disturbance can be seen for channel 5 at 150° and 180° azimuth. Channel 5 has a cen-
ter frequency of 26.24 GHz which is susceptible to frequencies used in communication links, of which there
are a few in and around the JOYCE site. That is why this disturbance is presumably due to RFI. As both of
these significant disturbances are within the K-band, they pose no threat to T-profile retrievals. Even H-
profiles retrievals, which only use zenith pointing channels in the K-band, are not affected by these disturb-
ances at 30° elevation. However they have to be taken into account and flagged when doing azimuth scans
which can be used for determining water vapor inhomogeneity.

Returning the focus to the V-band, channel 8 and 9, although used for T-profiling, are also sensitive to
humidity, as already discussed in Section 4.1. This can be seen in Figure 9, too. Between 220° and 60° azi-
muth, with a maximum at around 320° in north-west, this behavior leads to a higher probability of disturb-
ance and therefore to higher mean scan differences than in south-easterly directions during a whole year.
These disturbances could theoretically be due to horizontal inhomogeneities in water vapor in the atmos-
phere, but as already mentioned in Section 4.1, the mean impact of such conditions is low compared to the
far greater impact a small tilt of the instrument can cause. That is why it is far more probable that a small



tilt to the north-west and therefore a pointing error is the reason for these differences (compare with Section 4.2 and Figure 4, tilt here has therefore to be < 0.3°). Note that the HATPRO which measured the RFI in Figure 9 is not the same HATPRO as used Section 4.1. As already shown in Section 4.1, disturbances in

channel 8 and 9 have a low impact on T-profiling, especially when the disturbances in these channels are below 2.6 K (as seen in Figure 2), as it is the case here (analysis has shown that the probability of a scan difference of > 2.6 K is always 0 %, meaning there was no single azimuth scan which has more than a 2.6 K scan difference).

The channels in the K-band, especially channels 1–3, show a similar amplitude pattern as channels 8–9,

although the phases do not match up perfectly. Additional analysis of mean scan differences at 30° elevation has shown that the shift in phase is about 60° azimuth between K- and V-band. This shift is always observable, independent of weather condition, diurnal cycle and time of year. The other HATPRO – which for a 3 week period stood right next to the JOYCE HATPRO (and which was used for the measurements in Section 4.1) – does not show any phase shift in mean scan difference during that time. This strongly indicates that

internal uncertainties or misalignments within the JOYCE HATPRO instrument cause this mismatch in phase between K- and V-band channels.

Nevertheless, even though all of the aforementioned disturbances do not significantly affect T-profiling (as they predominantly occur within the K-band or within channel 8 and 9), they have to be monitored and assessed when installing a MWR at a new site, especially when azimuth scans are to be used to quantita-

tively analyze horizontal inhomogeneities of humidity.

## 5. Summary and conclusions

In this study, measurement uncertainties from HATPRO microwave radiometers and their impact on T-profiling have been analyzed. These measurement uncertainties included horizontal inhomogeneities of the atmosphere, pointing errors or tilts of the instrument, physical obstacles which are in the line of sight of the

MWR, and RFIs. The pointing errors and obstacles have been simulated with the help of a line-by-line RT model, while the instrument misalignments, horizontal inhomogeneities and an example of RFI have been analyzed through measurements on site at JOYCE.

Mean north-south TB differences at the same elevation angle during a 3 week period in summer 2022 can be mostly explained by a small tilt (about 0.3°) of the instrument and not by actual horizontal inhomogene-

ities. Therefore, before analyzing horizontal inhomogeneities, special care has to be made to align the instrument perfectly horizontal. Within the V-band, the largest mean differences of TBs in north- and south-facing scans have been observed in channels 8–10 at 10.2° and 19.2° elevation and are not exceeding 2.6 K, while for channels 11–14 they are always below 0.1 K. In order to achieve similar mean differences in TBs from actual horizontal inhomogeneities of water vapor from north- and south-facing scans in the V-band,

an average $\Delta IWV \approx 3.4$ kg m$^{-2}$ would be necessary which is highly unlikely and thus an instrument tilt is assumed. The impact of these measured mean north-south TB differences on retrieved T-profiles is small (median $\Delta$T-profile < 0.08 K), as the channels 8–10, which show the largest mean differences and standard deviations at various elevations, are only used in zenith for T-profile retrievals. Actual horizontal inhomogeneities in the retrieved T-profiles are represented in the percentiles range (25th/75th percentile ≤ |0.22 K|

beneath 3000 m).



Simulated pointing errors or tilts of the instrument up to ±1° only show a small impact on T-profiles. When using elevation scans in the T-profile retrievals, differences due to tilt do not exceed 0.1 K below 3000 m. When using zenith only observations, tilts of up to ±1° have almost no impact at all. In general, however, T-profile retrievals which make use of elevation scans are more accurate and reliable than retrievals which do not, especially in the lower 200 m and even when they have a tilt of 1°. The precise determination of the sources and magnitudes of tilts, whether originating from the set-up (external misalignment, such as the instrument's placement on a table) or from the instrument itself (internal misalignment, such as a misaligned mirror within the HATPRO), remains a subject of ongoing investigation and may constitute a future research endeavor.

Physical obstacles like trees, masts, buildings and mountains can have a strong impact on TB measurements and T-profiles, depending on their size, temperature and their distance to the radiometer location, especially at low elevations. Channels 8–10, which have the deepest penetration depths in the atmosphere, are most affected by simulated obstacles and can even "see" them from more than 10 km away at the lowest elevation angle of 5.4°, if they fill out the whole beam of the MWR in an inversion-free atmosphere. Channels 11–14 cannot reach as far into the atmosphere and can detect obstacles with ambient temperature at low elevations up to 3000–500 m away. When the temperature of the obstacles are 5 K above their surroundings, these distances for channels 11–14 increase to around 5500–1600 m at low elevations. In order for an obstacle to have a minimal impact on T-profiles of lower than 0.1 K, it has to be at least 600 m away. When the obstacle is 5 K warmer, this distance increases to at least 2700 m. Large obstacles like nearby mountains, which can also be seen in higher elevations, increase these distances further up to 4000 m in the worst case.

The impact of RFI on T-profiling – at least in our example – is negligible, when they occur around or near commonly used frequencies for communication links, which are usually situated within the K-band (mostly between 20–30 GHz). They can however negatively affect the analysis of K-band TBs in off-zenith directions which bear potential for deriving horizontal water vapor inhomogeneities.

In the following – with all these measurement uncertainties in mind – we will give some recommendations on how to properly set up a MWR. As a general rule, the operator needs to make sure that no obvious obstacles are near and around the scannable area of the MWR. If one locates possible obstacles, try to align the MWR in a way that there are no obvious obstacles in the preferred direction for elevation scans. While setting up the instrument, also make sure that the table on which the MWR is standing on is as level as possible, as even small tilts of under 0.5° can still cause a rather big influence on TB measurements in water vapor sensitive channels. After having done so (and after a recommended absolute calibration with liquid nitrogen) it is wise to initiate full azimuth scans at several elevations, similar to the 4-step method described in Section 4.4, for as long as possible (a few days at clear-sky conditions would be optimal). This allows the operator to identify all sorts of disturbances in all the different compass directions and elevations for all frequency channels, from nearby obstacles and RFIs to probable tilts of the instrument (see Figure 9 as an example). Directions with high disturbances should be avoided for obtaining retrieved products. In order to find out if a tilt comes from internal or external sources, one could set up an elevation scan at e.g. 30° (north-facing) and 150° (south-facing), so that the MWR will observe in the same direction. If TB measurement from these two scans are different, then there might be problem with the alignment inside the instrument.



Regarding on how strong various TB disturbances can affect measurements and profiling in more detail, more data in form of simulations and measurements are needed. An interesting aspect for further analysis could be a more in-depth analysis of how exactly the emissivity, temperature and size of an obstacle in line of sight of the radiometer would influence measurements. Another interesting aspect would be a more detailed simulation of horizontal inhomogeneities of water vapor and also temperature and how they affect T-profiling, especially in regard to pointing errors. With the help of mean azimuth scan differences, i.e. their amplitudes and phases, it is possible to determine the magnitude and direction of the instrument tilt. With that information it is theoretically possible to correct TB measurements in hindsight. By conducting further simulated experiments under controlled conditions, it will be possible to assess the potential benefits of retrospective corrections and optimize correction algorithms. These future simulations hold promise for enhancing the accuracy and reliability of MWR measurements, ultimately contributing to improved atmospheric observations. Finally, there remains a lack of comprehensive understanding regarding RFIs and their implications on ground-based MWR measurements, necessitating further investigation (see also WMO statements and guidelines from the Expert Team on Radio Frequency Coordination[7]).

### Code and data availability

The radiative transfer model used in this study can be found at https://doi.org/10.5281/zenodo.7990845. The raw MWR observations were processed with the software mwr_pro available at https://doi.org/10.5281/zenodo.7973552. Data from MWR measurements used in this study are available at the Institute for Geophysics and Meteorology of the University of Cologne.

### Author contribution

TB, UL and BP designed this study together. TB evaluated the data, produced the figures and wrote the manuscript with the help of UL and BP. UL is the author of the original radiative transfer model code which was altered by TB to fit the needs of this study.

### Competing interests

The corresponding author has declared that none of the authors have any competing interests.

### Acknowledgements

This project is funded by the project TDYN-PRO (Integration of Ground-based Thermodynamic Profilers into the DWD Forecasting System) within the funding line "Extramurale Forschung" of the German Weather Service DWD under the grant identifier 4819EMF02. The authors gratefully thank Annika Schomburg, Jasmin Vural, Moritz Löffler and Christine Knist from the German Weather Service for their support and helpful

---

[7] WMO Expert Team on Radio Frequency Coordination (ET-RFC): https://community.wmo.int/en/governance/commission-membership/commission-observation-infrastructure-and-information-systems-infcom/standing-committee-earth-observing-systems-and-monitoring-networks-sc/expert-team-radio-frequency-coordination-et-rfc



discussions. This research is embedded in CPEX-LAB (Cloud and Precipitation Exploration Laboratory within the Geoverbund ABC/J, http://www.cpex-lab.de) which includes the Jülich ObservatorY for Cloud Evolution (JOYCE) as a central measurement infrastructure. JOYCE is part of the European Research Infrastructure Consortium ACTRIS and is supported by the German Federal Ministry for Education and Research (BMBF) under the grant identifiers 01LK2001G and 01LK2002F. The paper has been motivated by collaborative concepts developed within the EU COST Action CA18235 "PROBE" (European Cooperation in Science and Technology), funding agency for research and innovation networks (http://www.cost.eu). Responsibility for the content of the publication lies with the authors.

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
