# Peer review of "Measurement uncertainties of scanning microwave radiometers and their influence on temperature profiling"

_EGUsphere, 2023_

## Referee Comment (RC2)

Comments on manuscript: "Measurement uncertainties of scanning microwave radiometers and their influence on temperature profiling" by Böck et al.
ID: egusphere-2023-1183

**Summary:**

The paper discusses the topic of ground-based scanning radiometers and associated uncertainty related to the measurements and temperature retrievals. Uncertainties discussed include horizontal inhomogeneity, radiometer tilt, obstacles in the scanning path, and RFI. The paper addresses these aspects with simulations and data from a field campaign.

**General comments:**

In my opinion the manuscript is a little bit confused in several aspects and would benefit from rethinking of some sections. I offer here some detailed comments on the parts that could use improvement and clarifications.

The main shortcoming of the paper as it stands is the assumption explicitly stated in line 315 that "*K-band channels do not play a role*" in temperature retrieval. This assumption leads to several inaccuracies in the approach and great confusion for the reader in the understanding of how clouds and vapor are treated in the retrievals. The main problem is due to the fact that channels 8 (51.26), 9 (52.28), and in less amount 10 (53.86) are indeed sensitive to both vapor and liquid water for which the K-band channels are needed. The authors should therefore address in better detail how they are accounting for vapor and clouds effects in channels 8-10 even if they only use the zenith measurements.

**Section 3.2 Retrieval method**: It is not clear how the forward model simulations are conducted and how the coefficients are derived. Are clouds included in the forward model simulations? If yes, how are they included? Same questions for the retrieval coefficients.

**Figure 1**: If I well understand Fig. 1 shows the TBs simulated and then inverted to derive temperature profiles. Are all channels (8-14) used in the scanning configuration? Of course, these simulations predict an improvement of the scanning configuration over the zenith because in the simulations all the shortcomings of real-world scanning are absent. Although it is true that in a perfect clear sky scenario of a perfectly homogeneous atmosphere in a perfectly flat and free horizon, scanning would improve the sensitivity of the temperature retrieval, the interpretation of measurements from scanning instruments is not always simple. Additionally, it is my impression that the excessive standard deviation of the zenith view retrievals in the first 200 m is due to the fact that the retrieval doesn't include surface temperature, pressure, and humidity. These quantities (available from the hatpro) are essential in the zenith configuration to constrain the first retrieval level where the channels have no sensitivity. It would be important to include those measurements when developing the retrieval coefficients.

**Figure 1:** Is the standard deviation of the differences a measure of accuracy? I don't think this is true. It is rather a measure of precision. Perhaps the RMS error between radiosondes and

retrievals can give us a better understanding of the two retrievals? In any case I think the label "*accuracy*" shouldn't apply here.

**Section 4.1**: it is stated in **line 196** that only clear sky cases are used to analyze the impact of horizontal inhomogeneity. However, in lines 220-230 and following the discussion of cloudy scenes is mentioned therefore it is not clear to the reader what is really being discussed here.

Same **in section 4.1.2.** Are the profiles of section 4.1.2 clear sky or all cases? If cloudy scenes are included in the retrievals but PWV and LWP are not retrieved there will be a mismatch between what channels 8-10 detect and what the other channels detect, potentially leading to incorrect results even using channels 8-10 in zenith mode. This because the signal from vapor and clouds is interpreted in the retrieval as a signal from temperature.

**Section 4.1.2**. I think this section requires more discussion because the intended meaning is not clear. If I scan the instrument both sides, I am going to retrieve temperature using both scans. In this case, even if the instrument is tilted, the average of the two brightness temperatures at the corresponding angles should take care of the bias introduced by the tilt. Therefore, the effect of the radiometer tilt will be introduced only if the radiometer is scanned on one side. For this and many other reasons *scanning on one side is never recommended*. This section should perhaps clarify this concept.

**Section 4.1.2.** Why not show the actual retrievals from the scanning configuration and the zenith configuration compared to the radiosondes during the JOYCE campaign? I think it will provide good information on how much improvement we can gain from scanning the radiometer from real world measurements rather than simulations.

**Section 4.2** Again the importance of scanning both sides should be stated.

**Figure 5:** Do retrievals in this figure use scanning data at all channels or only channels 11-14?

**Figure 5:** x-axis label should be RS-RET (K) – without *"elevation"*

**Line 343:** *"Nevertheless … pointing errors of up to ±1°"*. Again if I well understand, these results in Figure 5 are simulations. If these simulations are conducted scanning all channels (8-14) they are very difficult to implement in the real world because of the highly varying vapor and cloud fields that will require scanning the K-band channels as well.

**Section 4.3:** If this section is meant to be a guide for users on field installation in my opinion is not very practical. When in the field, it is hard to know the temperature of an obstacle located 1 km away and in table 2 there is not a direct connection with the height of the object. For example, an object located 3.5 km away needs to be at least 600 m tall to be detected at an elevation of 10.2 degrees (1st row, 3rd column in Table 2). Therefore, anything short of a small mountain or a very tall skyscraper won't be detected by the instrument at that elevation angle. In my opinion, the most direct question people face in the field is: *How far (minimum distance) from a XX m tall obstacle do I need to install the radiometer if I want to scan down to YY*

*degrees elevation?* The answer could be given as a table of which I draw a simplified example below. The paper could also come up with an approximate way to calculate that distance in the field for each channel without the need to run a radiative transfer code.

| Height of object (m) | Elevation | Minimum Distance (m) | | |
|---|---|---|---|---|
| | | CH1 | CH2 | CH3 |
| **10** Nearby antennas, power lines, small trees, 3-story buildings | 5.4 | 100 | ... | ... |
| | 10.2 | 55 | ... | ... |
| | 19.2 | 2 | ... | ... |
| | 30.0 | ... | ... | ... |
| **30** Power lines, medium trees, tall apartment buildings | 5.4 | 300 | ... | ... |
| | 10.2 | 16 | ... | ... |
| | 19.2 | 86 | ... | ... |
| | 30.0 | ... | ... | ... |
| **100** Cell tower, building towers, skyscrapers, surrounding hills | 5.4 | 11,000 | ... | ... |
| | 10.2 | 600 | ... | ... |
| | 19.2 | 300 | ... | ... |
| | 30.0 | ... | ... | ... |
| **300** Tall skyscrapers, surrounding mountains, cell towers, | 5.4 | ... | ... | ... |
| | 10.2 | ... | ... | ... |
| | 19.2 | ... | ... | ... |
| | 30.0 | ... | ... | ... |

---

## Author Comment (AC1)

**COMMENT 1:**

Ground-based microwave radiometers (MWRs) are used by many to provide measurements of the temperature profile and its evolution in the boundary layer. This paper builds upon work by other authors, exploring some important aspects that could lead to errors in the retrieved profiles. In particular, explore the role of horizontal inhomogeneities near the instrument, tilt of the radiometer, physical obstructions, and radio frequency interference (RFI).

My main question to the authors is simply: how does this paper add to the already extensive number of papers that address many of these topics? They reference the Meunier et al 2013 paper (but don't include any of those findings in their discussion here -- indeed, they ignore the possible uncertainties associated with beamwidth totally). There were also good papers by Han and Westwater (2000) and Liljegren (2000) that should have been referenced, and from which this paper should build. Regardless, the authors need to clearly state the new knowledge this paper is contributing to the field.

- First of all, thanks for your insightful comments. This paper tries to give a comprehensive overview of the measurement uncertainties of the newest state-of-the-art Gen5 HATPROs. Older papers do not describe all of these uncertainties (e.g. impact of obstacles or tilt) and when they do, it is often outdated information from older instruments. For network operation, these uncertainties are very important, that is why ACTRIS, E-PROFILE and the DWD need these up-to-date information.
  A passage has been added in Section 1 to make that more clear:

  - "For all that it is crucial to assess the uncertainties of state-of-the-art MWRs, as there is no comprehensive analysis and overview of uncertainties of these newest instruments yet. This study analyses measurement uncertainties by external sources, but uncertainties also encompass instrument uncertainties like biases, drifts, random noise and calibration errors which have been partly discussed in previous studies (Liljegren, 2002; Crewell & Löhnert, 2003; Maschwitz et al., 2013; Küchler et al., 2016) but are in need of updating."

- Liljegren (2002) and Han & Westwater (2000) references have been added to Section 1 and 2.
- Possible uncertainties associated with beam width are negligible small in the V-band with elevation angles >4° (<0.05K), according to Meunier et al. (2013). This is clearly stated at the end of Section 2. As there are no notable uncertainties associated with beam width in our case, there is no need to discuss this further in the paper.

I also found that this paper has a very informal tone about it; i.e., the language is a bit "loose". Many of the statements are repeated multiple times and could be better organized. There were multiple radiosonde (and retrieval coefficient) datasets used and at least two different MWRs; I suspect this does not impact the overall results at all, but it does add confusion.

- Repetition of certain important aspects are repeated multiple times on purpose to remind the reader, when needed. Nevertheless, we tried to make the language more precise in various sections.
- Section 3 already tries to make it clear where the data comes from and why.
- A new passage in Section 5 has been added to summarize all that:

  - "On one hand, the pointing errors and obstacles have been simulated with the help of a line-by-line RT model. The forward model implements radiosondes from RAO as input and the T-profile retrievals utilize coefficients from RAO as well. On the other hand, the instrument misalignments, horizontal inhomogeneities and an example of RFI have been analyzed through measurements on site at JOYCE, and T-profile retrievals from those measurements utilize coefficients from nearby De Bilt, Nether-
lands."

Also, the focus of the paper (from the title and abstract) was on the impact to the retrieved temperature profiles, and thus the inclusion of the discussion on the K band channels is distracting from the main message. And in many ways, it seemed that the K-band results were included as an afterthought, and not well organized. I would recommend that either the title changed and the K-band results be separated into their own (sub)sections, or that the K-band results be removed.

- Indeed, the main focus of the paper is the impact on the T-profiles (and hence the V-band channels), as it is stated multiple times. Nevertheless, the impact of RFI in the water vapor channels are briefly mentioned at the end of the abstract. The inclusion of the K-band channels throughout the paper is very brief and only complements our findings and is important for future research in how uncertainties could affect measurements on humidity advection/horizontal inhomogeneity of water vapor.
- Most radiometers (especially HATPRO-G5) do measure within the K-band, that is why it should be shortly introduced.
- We decided against changing the title of the paper, because it describes the content of the paper very well. We also decided to not get rid of all K-band results, as they are needed to paint a full picture. Removing the K-band would also result in losing our example we have of RFI. Some lines have been tweaked in Section 5 to make it more clear, that K-band findings do not play a role for T-profiling:
    - "The impact of RFI on T-profiling – at least in our example – is nonexistent, when they occur around or near commonly used frequencies for communication links, which are usually situated within the K-band (mostly between 20–30 GHz). These frequencies are not utilized in T-profile retrievals. However, RFI can negatively affect the analysis of observed TBs within the K-band in off-zenith directions, which bear the potential for deriving horizontal water vapor inhomogeneities."

  In the other Sections (4.4. and 4.2.) it is already clearly stated that K-band channels are not a focus and are therefore only briefly mentioned.

- We removed the description of the phase shift between K- and V-band in Section 4.4. though (line 479-486), as it is not really necessary within the scope of this paper, which hopefully reduces possible confusion for the reader.

I think that a key point that is being made here, but not explicitly stated, is the importance of always collecting elevation scans on both sides of the MWR. This allows the analyst to determine (from a sufficiently long dataset) the possible tilt of the instrument and the frequency of horizontal inhomogeneities. I think this point should be stated strongly, as I've observed many groups who believe it is sufficient to only collect elevation angles along one side of the radiometer.

- It is indeed better to collect elevation angles scans from both sides of the instrument during measurements, if this is possible. Oftentimes though it can be very difficult to scan down to 5° on both sides.
- That is why it is important to do 360° azimuth scans before doing regular measurements (which is stated in the paper in Section 4.4. and 5), in order to figure out possible tilts and inhomogeneities.
- Statements have been added (Section 4.2.2. and 5.) that collection of scans on both sides of the radiometer is recommended when possible, but that this also introduces some other challenges (see Section 4.2.2.):

- "In real-world scenarios, scanning on both sides of the radiometer – if possible – can mitigate pointing uncertainties to a certain degree, when T-profiles are retrieved from the average of such scans. However, this approach comes with problems, such as longer measurement times, the assumption of horizontal homogeneity of the atmosphere, and the assumption that elevation scans on both sides even out linearly."
- "If full azimuth scans are not feasible, at least elevation scans on both sides of the MWR are recommended, when there is the possibility to scan down to 5° elevation in both directions. Scanning in only one direction is sufficient though to retrieve accurate T-profiles, when the instrument is set up properly."

- For accurate T-profiling, scanning on one side of the instrument is sufficient though, if the instrument is set up properly. Possible (small) tilts of the instrument only have a very small impact (<0.1K for 1°) on T-profiles, as stated in Section 4.2.2.

The retrievals performed in this paper were done using a statistical method. Would have the results changed if a more accurate physical retrieval method was used instead? Line 79 indicates that the method of retrieval could matter (it might be useful to reference the Maahn et al. BAMS 2020 paper here for context -- Loehnert is a coauthor of that paper).

- For the purpose in this paper – a sensitivity study – it does not matter whether you use a statistical retrieval or a more accurate physical retrieval, as long as you use the same sort of retrieval all the time. Our results would be the same. We're mainly looking at the differences of the retrieved profiles, not at the absolute temperatures of the retrieved profiles. For this study, the use of a statistical retrieval was faster and easier to implement.
- Additionally, most operators do not use physical retrievals.
- A statement in the beginning of Section 5 has been added to address this issue:
  - "Regarding the retrieval method, a statistical approach has been employed. The utilization of alternative retrieval methods, such as a physically based one, would not yield a different outcome for this study."

Regarding obstructions: one of the more common setups that could affect these observations are power lines that are in the field of view of the radiometer. These lines clearly don't fill the entire field-of-view. Would the authors be able to provide any guidance on how far power lines would need to be away from the radiometer as to not impact the V-band observations?

- As the power lines are thin and would only fill out a very small portion of the beam, it is hard to say what impact they would have on V-band observations. It would depend on the thickness and number of powerlines in the field of view, and also on their temperature, which probably is higher than the ambient temperature. It is safe to say that avoiding power lines altogether or being as far away as possible (at least a few hundred meters) is the best bet. We cannot easily simulate power lines at different distances. Azimuth scans would reveal possible RFIs though.
- Statement for non beam-filling obstacles has been added in Section 4.3.:
  - "When encountering a small or slim obstacle that does not fill the entire beam width of the instrument, the resulting impact is generally less significant compared to larger obstacles. Simulating obstacles that do not completely fill the beam width of the MWR (such as power lines or lightning rods) poses challenges. Therefore, in our simulations, we focus on beam-filling obstacles for which a minimum distance can be determined at which they do not interfere with the measurements anymore. Our simulations of such beam-filling obstacles…"

Equations 2 and 3: why is the retrieved temperature a function of both frequency and height? I think you can remove the nu from the left side of both of those equations.

- Has been changed.

Line 230: Aren't rapid changes in zenith radiance observations also a measure of the horizontal inhomogeneity? Is there a way to look at the variability of the zenith radiance observations over time (and the trend of the magnitude of these radiance obs over time) to estimate the possible level of horizontal inhomogeneity?

- Yes, changes in zenith radiance are also a measure of horizontal inhomogeneities. In order to estimate the level of horizontal inhomogeneity reliably, elevations scans are needed though. Estimates on horizontal inhomogeneity only using zenith measurements are rather unprecise, as it only concerns the variability over the instrument, but does not include horizontal gradients.

Lines 315-320: (if you decide to keep the K-band material and reorganize it): the magnitude of the impact on the K-band as described in the text does not seem to match with the magnitudes shown in Fig 4.

- Has been checked. Description in the text does match with the magnitudes shown in Fig. 4.

You make the point a few times in different places that the impact of obstacles depends on the vertical structure of the temperature profile; namely, that there is a smaller impact when there is an inversion. This really should be discussed a bit more to explain why this is.

- Explanation has been slightly extended in Section 4.3.1:
  - "For an ambient temperature obstacle, the $\Delta TB$ can even become negative when there is an inversion, meaning that the ambient temperature obstacle near the colder surface blocks the MWR from observing warmer atmospheric layers above and beyond the obstacle."

This paper would be markedly stronger if a plot showing the changes in Tb and the associated changes in total optical depth for small elevation differences was shown. I've included such a figure here for a mid-day, midcontinental radiosonde with IWV=43 kg/m2, for small elevation angle changed around 5.4 deg. In particular, note that the change in the optical depth is constant, in a fraction changed sense, as the elevation angle changes. This also demonstrates why there is little impact for channels 12, 13, and 14 -- the optical depth is already very large.

[Figure]

- The plots you show are interesting, and the information in them is complementary to our results. But we think that they do not show anything new. In our opinion, the information already given in the paper is enough, as the analysis of changes in optical depth due to elevation is not a focus in this paper, but rather the impact of that for sensing obstacles.
- It is stated in the paper multiple times (Section 2, 4.1., 4.3.) that channels 8-10 are optically thinner than channels 11-14, which are optically thick and therefore do not penetrate the atmosphere very far. The results in Section 4.1. and 4.2. also make clear that as a result the optically thick channels 11-14 do not sense horizontal inhomogeneities.
- Table 2 also demonstrates the various penetration depths for different channels depending on the elevation angles.

Line 440: Clear sky scenes also have very little temporal variation in the observed Tb values. So not only should the mean LWP be small, but the standard deviation of the LWP should be small also.

- Indeed. New filtering has been added in Section 4.4.:
  - "If the mean LWP of a 30 minute interval before and after one scan is below 10 g m$^{-2}$ and its standard deviation below 4 g m$^{-2}$, then it is most likely clear-sky."

Line 452: Earlier, you made the point that obstacles needed to been many hundreds of meters away; yet this lightning rod is only 5 m away and has no impact. I suspect this is because the beamwidth, together with the 10-deg azimuth sampling, resulted in this rod not being in the V-band's field-of-view. Is that correct? Regardless, the way this section is worded conflicts with what you wrote earlier in the paper about obstructions.

- Paragraph has been added in Section 4.4. which addresses this phenomenon and explains better why the obstacle can only be seen in the K-band but not in the V-band:
  - "The lightning rod is not detected in the V-band channels primarily due to the narrower beam width of the V-band receiver. This is mostly attributed to the combination of the 10° azimuth step and the rod's slim profile, which results in the rod being predominantly positioned outside the field of view. Although it is possible that parts

of the rod may be partially within the field of view, the limited coverage within the field of view is not substantial enough to have a noticeable impact on the measurements."

Line 475: here is an example of the loose writing. You've already stated that off-zenith observations from channels 8 and 9 are not used in the temperature retrievals, so RFI disturbances in these channels at off-zenith angles should ZERO impact on the temperature retrievals (not negligible).

- Has been fixed and shortened:
    - "As already discussed in Section 4.1, off-zenith disturbances in channel 8 and 9 have no impact on T-profiling."

Line 485: I really don't understand how internal uncertainties / misalignments would cause these results. This needs to be explained, or (as is my preference) the K-band results removed.

- We are also not sure what causes this or how it is caused, it's only speculation. But the error source has to be "inside" the instrument, as external factors can be excluded. We removed this paragraph (line 479-486 in the original manuscript), as it is not crucial for T-profiling.

Line 500: this is a great place to emphasize the need to have matching elevation scans on both sides of the radiometer

- Initial azimuth scans, as suggested a few lines earlier, are a good method to detect disturbances, when feasible. Statement has been added, if not:
    - "If full azimuth scans are not feasible, at least elevation scans on both sides of the MWR are recommended, when there is the possibility to scan down to 5° elevation in both directions. During regular operation, scanning in only one direction is sufficient though to retrieve accurate T-profiles, when the instrument is set up properly."

Line 511: Are the errors in retrieved temperature profile small because the errors in the observed Tbs are somewhat offsetting because you are using data from both sides of the radiometer? I think yes.

- We're talking about simulated pointing errors here, so there's no data from both sides of the radiometer. Just a simulation at a certain elevation and another simulation at a slightly different elevation. Then the differences of these simulations and their influence on T-profiles are described. So there is no "offsetting" taking place.

---

## Author Comment (AC2)

**COMMENT 2:**

Summary:
The paper discusses the topic of ground-based scanning radiometers and associated uncertainty related to the measurements and temperature retrievals. Uncertainties discussed include horizontal inhomogeneity, radiometer tilt, obstacles in the scanning path, and RFI. The paper addresses these aspects with simulations and data from a field campaign.

General comments:
In my opinion the manuscript is a little bit confused in several aspects and would benefit from rethinking of some sections. I offer here some detailed comments on the parts that could use improvement and clarifications.

The main shortcoming of the paper as it stands is the assumption explicitly stated in line 315 that "K-band channels do not play a role" in temperature retrieval. This assumption leads to several inaccuracies in the approach and great confusion for the reader in the understanding of how clouds and vapor are treated in the retrievals. The main problem is due to the fact that channels 8 (51.26), 9 (52.28), and in less amount 10 (53.86) are indeed sensitive to both vapor and liquid water for which the K-band channels are needed. The authors should therefore address in better detail how they are accounting for vapor and clouds effects in channels 8-10 even if they only use the zenith measurements.

- Thank you for that comment. We are aware that the channels 8-10 (situated in the V-band) are also sensitive to water vapor and liquid water, and not only oxygen. However, the retrievals include all these contributions. No observations in the K-band (channels 1-7) are needed to create retrievals for T-profiles.
- We are accounting for water vapor [via Turner et al. (2009) Liljegren et al. (2005)] and also clouds (via Liebe et al., 1993) in the statistical retrieval (multivariate regressions). The forward RT model does not need to take clouds into account, as our analyses in this paper which use the RT model are for cloud-free scenarios. We addressed that in Section 3.1. and 3.2. and added more details. You can see these added details in the following comment of yours.
- For a better general understanding: it is stated in Section 3.2. that the retrieval for T-profiles only uses the V-band channels and that for all T-profile retrievals in this paper, which make use of elevation scans, only channels 11-14 are utilized with elevation scans, whereas channels 8-10 are always only utilized in zenith.

Section 3.2 Retrieval method: It is not clear how the forward model simulations are conducted and how the coefficients are derived. Are clouds included in the forward model simulations? If yes, how are they included? Same questions for the retrieval coefficients.

- Section 3.1. tries to explain how the forward model simulations work (we tweaked the description to reduce confusion of what is included):
  - "…non-scattering radiative transfer simulations are carried out based on Simmer (1994). Gaseous absorption is calculated according to Rosenkranz (1998), whereby the water vapor continuum is modified according to Turner et al. (2009) and the 22 GHz water vapor line width is modified according to Liljegren et al. (2005). This model depicts the energy transfer in form of electromagnetic radiation through the atmosphere via absorption and emission by gas molecules and it is modified here to simulate physical obstacles at different distances and elevation angles and to assess pointing errors in a cloud-free atmosphere."

- And Section 3.2. explains in detail how the coefficient of the retrievals are derived. There, clouds are taken into account (via Liebe et al., 1993), so that we can use the retrievals also for cloudy conditions. We added that statement to the text:

- "The applied coefficients of this retrieval model incorporate absorption and emission from liquid water (Liebe et al., 1993), in addition to water vapor and oxygen."

Figure 1: If I well understand Fig. 1 shows the TBs simulated and then inverted to derive temperature profiles. Are all channels (8-14) used in the scanning configuration? Of course, these simulations predict an improvement of the scanning configuration over the zenith because in the simulations all the shortcomings of real-world scanning are absent. Although it is true that in a perfect clear sky scenario of a perfectly homogeneous atmosphere in a perfectly flat and free horizon, scanning would improve the sensitivity of the temperature retrieval, the interpretation of measurements from scanning instruments is not always simple. Additionally, it is my impression that the excessive standard deviation of the zenith view retrievals in the first 200 m is due to the fact that the retrieval doesn't include surface temperature, pressure, and humidity. These quantities (available from the hatpro) are essential in the zenith configuration to constrain the first retrieval level where the channels have no sensitivity. It would be important to include those measurements when developing the retrieval coefficients.

- As stated in the retrieval section, only channels 11-14 are used with all elevations angles. Channels 8-10 are always only used in zenith. This is true for the whole study.
- Indeed, the retrieval does not use the surface temperature, but are in-situ surface temperatures representative for surroundings? Oftentimes not, e.g. heating from the roof can falsify these measurements. But in most cases, when there are no obstacles or RFI, T-retrievals which make use of elevation scans will always be more precise (especially in the lowest 1000m of the troposphere), and this is the most important fact, regardless if surface temperatures are included or not. This was also already shown in Crewell & Löhnert (2007).

Figure 1: Is the standard deviation of the differences a measure of accuracy? I don't think this is true. It is rather a measure of precision. Perhaps the RMS error between radiosondes and retrievals can give us a better understanding of the two retrievals? In any case I think the label "accuracy" shouldn't apply here.

- As there is no bias to speak of, we can use standard deviation instead of RMS, they tell the same thing.
- The label "accuracy" has been changed in all sections with "precision".

Section 4.1: it is stated in line 196 that only clear sky cases are used to analyze the impact of horizontal inhomogeneity. However, in lines 220-230 and following the discussion of cloudy scenes is mentioned therefore it is not clear to the reader what is really being discussed here. Same in section 4.1.2. Are the profiles of section 4.1.2 clear sky or all cases? If cloudy scenes are included in the retrievals but PWV and LWP are not retrieved there will be a mismatch between what channels 8-10 detect and what the other channels detect, potentially leading to incorrect results even using channels 8-10 in zenith mode. This because the signal from vapor and clouds is interpreted in the retrieval as a signal from temperature.

- Line added that will hopefully make it clearer that clear and cloudy sky cases are analyzed:
  - "To exclude the influence of clouds when needed during analysis, …"
- In Section 4.1.2 both clear and cloudy cases are included and discussed and labeled as such. We wanted to include cloudy cases, because this gives a more realistic/complete view on the impact of horizontal inhomogeneities.
- There is no mismatch. The signal that stems from water vapor and liquid clouds does not affect the quality of the temperature profile as these contributions are also already accounted for in the retrieval development. PWV and LWP do not need to be retrieved separately.

Section 4.1.2. I think this section requires more discussion because the intended meaning is not clear. If I scan the instrument both sides, I am going to retrieve temperature using both scans. In this case, even if the instrument is tilted, the average of the two brightness temperatures at the corresponding angles should take care of the bias introduced by the tilt. Therefore, the effect of the radiometer tilt will be introduced only if the radiometer is scanned on one side. For this and many other reasons scanning on one side is never recommended. This section should perhaps clarify this concept.

- In Section 4.1.2. the MWR scanned on both sides, but for the retrieval first only one side was taken into account, which was then compared to the other side. There is no averaging taking place between north- and south-facing scans.
- The same is true for the simulation in Section 4.2.1. The simulations only ever assume scanning on one side of the radiometer. But a paragraph has been added here to clarify what it would mean if we would average scans on both sides:
    - "In real-world scenarios, scanning on both sides of the radiometer – if possible – can mitigate pointing uncertainties to a certain degree, when T-profiles are retrieved from the average of such scans. However, this approach comes with problems, such as longer measurement times, the assumption of horizontal homogeneity of the atmosphere, and the assumption that elevation scans on both sides even out linearly."

- Please consider that oftentimes, it is not possible to scan down to 5° on both sides of the MWR.

Section 4.1.2. Why not show the actual retrievals from the scanning configuration and the zenith configuration compared to the radiosondes during the JOYCE campaign? I think it will provide good information on how much improvement we can gain from scanning the radiometer from real world measurements rather than simulations.

- We guess you are referring to Section 4.2.2. and the FESSTVaL campaign here? For the observations at JOYCE we did not have co-located radiosonde observations.
- It is long since known that elevation scans do improve the T-profiles (Crewell & Löhnert, 2007). We used this as a motivation why we primarily analyze elevation scans. Fig. 1 shows that improvement..
- The blue line from Fig.1 represents the right plot from Fig. 5. Retrievals are made from calculated TBs derived from radiosondes from RAO. We added a statement in Section 4.2.2., to make the connection between Fig. 1 and Fig. 5 more clear:
    - "… (compare the standard deviations from Figure 5 with Figure 1; the blue line in Figure 1 corresponds to the content displayed in the right plot of Figure 5 when there is no tilt)."

Section 4.2 Again the importance of scanning both sides should be stated.
- Has been added (see two comments above).

Figure 5: Do retrievals in this figure use scanning data at all channels or only channels 11-14?
- Only channels 11-14, as it is always the case in this study (stated in Section 3.2.)

Figure 5: x-axis label should be RS-RET (K) – without "elevation"
- Has been fixed.

Line 343: "Nevertheless … pointing errors of up to ±1°". Again if I well understand, these results in Figure 5 are simulations. If these simulations are conducted scanning all channels (8-14) they

are very difficult to implement in the real world because of the highly varying vapor and cloud fields that will require scanning the K-band channels as well.

- The TBs have been simulated or calculated from real-world radiosondes. And then the T-profiles have been retrieved from those TBs. As stated in previous comments, retrievals use only channels 11-14 with all elevations and channels 8-10 only in zenith (see Section 3.2.). K-band is not used at all, and retrievals also never use K-bands with elevations, as this would not make much sense. K-band channels are optically much too thin to make use of elevation scans. Only optically thick channels, like in the V-band (oxygen absorption) are suitable for elevation scans, as we reach different layers/heights of the atmosphere by doing such scans in these channels.

Section 4.3: If this section is meant to be a guide for users on field installation in my opinion is not very practical. When in the field, it is hard to know the temperature of an obstacle located 1km away and in table 2 there is not a direct connection with the height of the object. For example, an object located 3.5 km away needs to be at least 600 m tall to be detected at an elevation of 10.2 degrees (1st row, 3rd column in Table 2). Therefore, anything short of a small mountain or a very tall skyscraper won't be detected by the instrument at that elevation angle. In my opinion, the most direct question people face in the field is: How far (minimum distance) from a XX m tall obstacle do I need to install the radiometer if I want to scan down to YY degrees elevation? The answer could be given as a table of which I draw a simplified example below. The paper could also come up with an approximate way to calculate that distance in the field for each channel without the need to run a radiative transfer code.

- Table 2 already gives an overview of how far an obstacle needs to be away where it has no more influence on the observation. It does indeed not state how tall this obstacle would need to be, but this would be easy to find out via trigonometry. The end of section 4.3.2. also gives rough examples of the connection of obstacle distances and heights.
  But you're right that users wouldn't really know the temperature of an obstacle, which makes it harder to estimate minimum distances. Another factor is that the beam radius gets larger and larger the further away it is from the instrument, meaning that obstacles far away will probably not fill out the entire beam.
  That is why a full azimuth scan at different elevations at a new location is always advisable in order to detect such obstacles and if they may pose problems. Such a statement has been added in Section 5:
    - "In practical scenarios, accurately estimating the temperature and appropriate distance of an obstacle can pose challenging for the operator, particularly when the obstacle occupies only a small portion of the instrument's beam. This scanning method proves invaluable in such cases."

- The inclusion of such a table, where minimum distances of obstacles and their corresponding minimum height are written down, is an interesting idea but we are not sure if it would really benefit the reader. For example: channel 13 has a maximum penetration depth of ~600m at 5° elevation, meaning that an obstacle which is 600m or more away from the radiometer does not get noticed anymore. At 600m distance, the obstacle would need to stand at least ~57m tall, to be in the line of sight. But it could also be much taller than that (or smaller), it doesn't matter, because it is already too far way. If the obstacle is, let's say, only 40m tall and only 200m away, it would have an impact in channel 13. So including a table which only states minimum distances and their corresponding height would only paint an incomplete picture. We would have to include a lot of examples at various distances, which would render such a table rather confusing and not really helpful.
  Only knowing the maximum penetration depth of a channel should be enough for the operator. If there is a tall obstacle somewhere in the line of sight, the operator, knowing the maximum penetration depth of a channel at a certain elevation, can then calculate if the obstacle is too tall/big or not.